# X Chromosome Inactivation in Carriers of Fabry Disease: Review and Meta-Analysis

**DOI:** 10.3390/ijms22147663

**Published:** 2021-07-17

**Authors:** Emanuela Viggiano, Luisa Politano

**Affiliations:** 1Department of Prevention, UOC Hygiene Service and Public Health, ASL Roma 2, 00142 Rome, Italy; 2Cardiomyology and Medical Genetics, Department of Experimental Medicine, Luigi Vanvitelli University, 80138 Naples, Italy

**Keywords:** Fabry disease, carriers, X chromosome inactivation

## Abstract

Anderson-Fabry disease is an X-linked inborn error of glycosphingolipid catabolism caused by a deficiency of α-galactosidase A. The incidence ranges between 1: 40,000 and 1:117,000 of live male births. In Italy, an estimate of incidence is available only for the north-western Italy, where it is of approximately 1:4000. Clinical symptoms include angiokeratomas, corneal dystrophy, and neurological, cardiac and kidney involvement. The prevalence of symptomatic female carriers is about 70%, and in some cases, they can exhibit a severe phenotype. Previous studies suggest a correlation between skewed X chromosome inactivation and symptoms in carriers of X-linked disease, including Fabry disease. In this review, we briefly summarize the disease, focusing on the clinical symptoms of carriers and analysis of the studies so far published in regards to X chromosome inactivation pattern, and manifesting Fabry carriers. Out of 151 records identified, only five reported the correlation between the analysis of XCI in leukocytes and the related phenotype in Fabry carriers, in particular evaluating the Mainz Severity Score Index or cardiac involvement. The meta-analysis did not show any correlation between MSSI or cardiac involvement and skewed XCI, likely because the analysis of XCI in leukocytes is not useful for predicting the phenotype in Fabry carriers.

## 1. Anderson-Fabry Disease

Anderson-Fabry disease (OMIM #301500) is a progressive X-linked inborn error of glycosphingolipid catabolism caused by a deficiency of α-galactosidase A (α-gal), belonging to the family of lysosomal exoglycosidases. The incidence ranges between 1:40,000 and 1:117,000 [1,2] live male births. In Italy, an estimate of incidence is available only for north-western Italy, where it is approximately 1:4000 [3]. The *GLA* gene is located on Xq22.1 and consists of 7 exons, encoding the α-gal protein. It is a polypeptide of 429 amino acids, while the mature protein is a glycoprotein of about 100 kDa, with a homodimer structure [4]. So far, more than 900 variants have been reported in the Human Gene Mutation Database (http://www.hgmd.cf.ac.uk/ac/gene.php?gene=GLA, accessed on 16 July 2021), about 69% of which are missense mutations, 17% deletions, 5% splicing and 5% insertions or duplications. The pathogenetic mechanisms of Fabry disease are not completely clear. The enzyme deficiency determines a progressive accumulation of glycolipids, in particular globotriaosylceramide (Gb3) in the lysosomes and cytosol of cells in several tissues, predominantly in cardiovascular, peripheral and central nervous systems, and kidney [5,6,7]. The onset of symptoms and the phenotype depend on the residual enzyme activity; in particular, a level less than 1% of normal activity leads to the classical form, and levels between 1 to 30% to atypical forms [8,9,10,11].

### 1.1. Inheritance

Fabry’s disease is inherited in an X-linked manner. The heterozygous mothers have a 50% chance of passing the defective gene to all offspring at each conception. The sons who inherit the defective gene will have Fabry’s disease; the daughters, once thought to be asymptomatic carriers, may instead develop disease manifestations from mild to severe [12]. Though a positive family history is a strong indicator of Fabry’s disease, de novo mutations have been documented, and so the absence of a family history does not rule it out.

### 1.2. Pathophysiology

The functional defect of α-gal leads to a progressive accumulation of Gb3 in all body cells containing lysosomes, including vascular endothelium and smooth muscle cells, cardiomyocytes in the left ventricle and atrium, valvular fibroblasts, endothelial cells and vascular smooth muscle cells [13]. However, Gb3 storage cannot alone explain the absence of complications in newborns [14], the absence of correlation between the severity of clinical symptoms and the level of Gb3 in plasma or urine [15,16], symptoms in female carriers despite the presence of residual α-gal enzyme and the intra-familial phenotypic variability despite the same genetic mutation [17,18]. Moreover, the contribution of the stored material to the increase in cardiac mass was limited to about 1–2% [7]. For those reasons, some researchers argue that the increased levels of Gb3 can trigger other pathogenetic mechanisms. First, an increased level of globotriaosylsphingosine (Lyso-Gb3) is observed in the plasma of Fabry patients, including female carriers [19]. Lyso-Gb3 is involved in the vasculopathy, as it induces the proliferation of smooth muscle cells that is associated with hypertrophy and myocardial arterial walls and fibrosis, in vitro [20,21,22]. The storage of Gb3 induces an excessive production of reactive oxygen, thereby increasing oxidative stress [23]. In particular, the increase in radical oxygen production and the decrease of anti-oxidants, such as glutathione and superoxide, cause an impairment of the myofilaments, altering contractility and distensibility, and favoring apoptosis at the cardiac level [24,25]. Gb3 also up-regulates the expression of adherence molecules in vascular endothelium [26]. Other data indicate that Gb3 may cause the release of pro-inflammatory cytokines, especially in dendritic cells and monocytes [27], since inflammatory mediators (such as TNF-α, IL 1β, IL-6) are increased in the plasma of patients with Fabry [27,28]. Moreover, GB3 would activate the innate response, through binding the TLR4 receptor on immune cells, which increases the release of inflammatory mediators from peripheral blood mononuclear cells [28]. The role of inflammation is also supported by the endomyocardial biopsy that shows an increase in inflammatory macrophages in the tissue [29,30]. The inflammatory response can also be triggered by the vascular alteration. In fact, vascular remodeling could reduce the arterial compliance, determining upregulation of the renin-angiotensin system and the increase of angiotensin 1 and 2 in endothelial cells [31,32], which trigger the inflammation [31,33]. The consequence of inflammation is the release of pro-thrombotic factors in the vessels, increasing the risk of ischemia by increasing the extracellular matrix deposition and fibrosis [31,34].

These data suggest that the Gb3 storage may act by triggering a cascade of pathophysiological processes leading to structural cellular changes, tissue defects, and—over time—to organ failure.

### 1.3. Clinical Presentation

The classical form of Fabry disease develops in childhood or adolescence, usually with endothelial dysfunction leading to angiokeratomas, and corneal dystrophy, that represent the physical stigmata of the disease [35]. The progression of the disease involves the peripheral and autonomic nervous systems leading to paraesthesia, neuropathic pain; in particular “Fabry crises” are described as pain in the extremities which radiates proximally. Other symptoms depend by the involvement of gastrointestinal systems, in particular abdominal pain, diarrhea or constipation, especially in female carriers [36], and hypohidrosis; less frequently hyperhidrosis. Moreover, patients can present cerebrovascular involvement, in particular transient ischemic attacks (TIA) and stroke.

Finally, kidneys and heart are affected, and, when untreated, patients can die from heart and/or kidney failure in the fourth or fifth decade of life [5]. The renal involvement in Fabry’s has been known since the original reports by Anderson and Fabry in the late nineteenth century. It is likely to start at an early age, around the age of 22, and to be more severe in patients with α-gal enzyme less than 1% than in those with detectable enzyme levels, in whom renal involvement begins late in life. Urinary concentration defects, polyuria, proteinuria, and chronic renal insufficiency are known as clinical renal manifestations of Fabry’s disease. Concentration defects are the earliest functional manifestation. Proteinuria may begin in the teens, becoming more common in the second–third decade. Twenty-five percent of patients may progress to chronic renal insufficiency [37].

Facial dysmorphisms are reported in male Fabry patients, characterized by peri-orbital fullness, prominent supra-orbital ridges, large bitemporal width, ptosis, broad nasal base, bulbous nasal tip, full lips and coarse features [38,39]. The identification of deposits of Gb3 at the reproductive tract level has suggested an impairment of male gonadal function in Fabry disease. Papaxanthos–Roche et al. have recently shown that patients with FD might have a detrimental effect on semen characteristics, but the reproductive function is only slightly diminished [40]. The impact of Fabry Disease on reproductive fitness was studied by Laney et al. on a large, multi-centered population (*n* = 376) of individuals, both males and females with FD. They conclude that in this large multicenter sample, patients with FD did not exhibit reduced reproductive fitness [41]. However, previous studies [42] have reported azoospermia as a possible complication of Fabry disease. A routine sperm analysis in the follow-up of young patients with Fabry disease was recommended as far as sperm cryopreservation.

The later-onset atypical forms are usually more benign, because they involve a more restricted number of organs, usually limited to the kidneys, heart or nervous system [1].

#### Cardiac Manifestations

Cardiac involvement in Fabry disease is frequent in male and in heterozygote females [43,44]. The progression of cardiomyopathy is determined by the involvement of cardiac muscle, and the conduction and vascular systems.

In particular, at the cardiac muscle level, patients with Fabry usually show concentric and non-obstructive left ventricular hypertrophy (LVH). Sometimes the cardiomyopathy in Fabry patients mimics the hypertrophic cardiomyopathy due to sarcomeric genes mutations, particularly when isolated. *MYH7* gene-encoding β-cardiac myosin heavy chain (MHC) and *MYBPC3*, which encodes myosin binding protein C, account for more than 50% of HCM patients with pathogenic variants [45,46,47]. Other sarcomere protein genes causing HCM include cardiac troponin T (*TNNT2*), cardiac troponin I (*TNNI3*), α-tropomyosin (*TPM1*), myosin regulatory light chain 2 (*MYL2*), myosin essential light chain (*MYL3*), and actin (*ACTC1*) [48,49]. However, the global ejection fraction is preserved [50]. Other typical findings in Fabry cardiomyopathy are prominent papillary muscles [43,44] and a preserved global ejection fraction combined with early stages of diastolic dysfunction [51,52]. The onset of cardiomyopathy usually occurs in males aged >30 years and in females aged >40 years. Valvular fibrosis leading to valvular abnormalities frequently occurs; however, the involvement is generally mild and clinically insignificant. Conduction system abnormalities like short PR interval, atrium-ventricular blocks, supraventricular and ventricular arrhythmias are also reported [53]. Myocardial ischemia, frequently observed, has in most instances a functional origin due to endothelial dysfunction of coronary arteries and to the increased oxygen demand of hypertrophic myocardium [51].

The end-stage Fabry cardiomyopathy is characterized by intramural replacement fibrosis limited to the basal postero-lateral wall of the left ventricle [43]

The most common diagnostic tools to assess cardiac functions in Fabry’s are electrocardiogram and Holter monitoring for conduction abnormalities, echocardiography and magnetic resonance imaging (MRI) for the myocardial mass assessment. Echocardiography, which is widely available and easily applicable, shows the early stages of the disease, the typical images of prominent papillary muscles as well as of a thickened interventricular septum and hypertrophy of the left ventricle lateral wall [54]. Echocardiography is also the method of choice to monitor treatment effects. However, the non-invasive gold-standard to detect myocardial fibrosis is late gadolinium-enhanced MRI [55].

### 1.4. Genotype-Phenotype Correlation

As in other inherited metabolic diseases, the phenotype usually depends on the residual enzyme activity, which in turn depends on the type of mutation [56]. In general, mutations, that cause less than 1% of enzyme activity, lead to the classical form, while those causing between 1–30% of normal activity, lead to atypical forms [8,9,10,11].

In the Fabry Outcome Survey (FOS), which included 545 patients belonging to 157 families from nine European countries, a highly significant positive correlation was found between the age at entry into FOS and the FOS Severity Index as well as between the age at entry into FOS and the number of affected organs (*p* < 0.001) in males with *GLA* missense mutations, irrespective of whether the change in the amino acid side chain predicted in the α-gal protein was classified as a conservative or non-conservative change [57].

However, the analysis of genotype–phenotype correlations in Fabry disease is complicated by a number of factors, such as the high proportion of private mutations, the large intra and inter-familial phenotypic heterogeneity associated with the same mutation, and disease-related complications observed with high prevalence in the general population [58,59,60,61,62,63].

### 1.5. Treatment

Comprehensive and timely treatment of adult patients with Fabry disease are directed toward prevention of further progression to irreversible tissue damage and organ failure. Care should include enzyme replacement therapy (ERT) and adjunctive therapies to treat symptoms that arise due to tissue injury and prevent non-specific progression of the disease [64].

Currently, two different forms of ERT are available; agalsidase-alfa (Replagal, Takeda), produced in human fibroblasts and registered at a dose of 0.2 mg/kg biweekly, and agalsidase-beta (Fabrazyme, Sanofi Genzyme), produced in Chinese hamster ovary cells and registered at a dose of 1.0 mg/kg biweekly. Long-term clinical studies have shown a small but significant effect of ERT on cardiovascular and renal complication rate, with some superiority of the higher dosed agalsidase-beta compared to agalsidase-alfa [64,65]. Especially loss of renal function, occurring in the vast majority of male patients with classic Fabry, is attenuated by ERT [66]. These clinical benefits were mainly observed in patients who started ERT before the presence of irreversible organ damage [67,68]. For patients with missense mutations that result in a mutant protein with normal α-gal catalytic activity and reduced protein stability, the use of a pharmacological chaperone is indicated, given its ability to bind and stabilize protein specifically [69]. In particular, the pharmacological chaperone 1-deoxy-galactonojirimycin (DGJ, Amigal or Migalastat) Migalastat (commercial name Galafold™) is currently the only oral treatment for Fabry disease approved by the US Food Drug Administration (FDA) and European Medicines Agency (EMA). Migalastat is able to stabilize α-gal, prolonging the half-life of α-gal in vivo, both in mouse models and in humans and leads to an improved clearance of Gb3 [70,71,72,73], but it also inhibits α-gal. For this reason, the therapy is intermittent, and consists in two phases: in the first phase Migalastat is administered, inhibiting and stabilizing α-gal, while in a second phase α-gal is activated in absence of the drug [71], A useful database with predictive tool for mutations, Fabry_CEP, can be used to evaluate the responsiveness of GLA mutations to Migalastat [74]. Migalastat can be used in synergy with ERT, either co-administrating both drugs intravenously or one orally (Migalastat) and the other intravenously (recombinant enzyme) [75,76].

Treatments currently under evaluation in preclinical trials are second generation ERTs (Pegunigalsidase-alfa, Moss-aGal), substrate reduction therapies (Venglustat and Lucerastat), mRNA and gene-based therapy [77]. The follow-up assessments to evaluate treatment responses should ideally be supervised by a physician experienced in the management of patients with Fabry disease, with input from sub-specialists who also have Fabry disease experience as part of a multidisciplinary clinical team that includes neurologists, nephrologists, cardiologists, clinical geneticists, genetic counsellors and psychologists [78]

## 2. Female Carriers of Fabry Disease

Given the X-linked pattern of inheritance, clinical manifestations of Fabry disease in heterozygous females, as obligate carriers, have long been considered rare or mild. However, the prevalence of symptomatic female carriers is estimated about 70% [79,80], with phenotypes ranging from very mild to severe cases [81,82,83]. Moreover, data in literature indicate that females are affected much more commonly than previously believed [84].

Fabry carriers usually show late onset of symptoms, slower progression of the disease and longer life expectancy, estimated to be around 70 years, compared to 50–55 for male patients [85,86]. Neuropathic pain is reported in about 10% of carriers, usually intermittently [87] and often misdiagnosed as polyarthritis or polyarteritis nodosa; cornea verticillata is present in about 72% of girls, while hearing impairment is reported in about 33% of cases [88,89]. Other reported symptoms in about 33% of cases are renal involvement and proteinuria [90,91], requiring dialysis or kidney transplantation in 10% of cases [92]. According to the Fabry Outcome Survey (FOS) [93], 65% of affected females develop cardiac involvement, including cardiac ischemia of microvascular origin, hypertrophic cardiomyopathy and arrhythmias (atrio-ventricular block, tachyarrhythmias and ST-segment/T-wave abnormalities) that may require pacemaker implantation [94,95,96]. Cardiac involvement correlates with age; in particular, Kampmann et al. [97] reported the presence of cardiomyopathy in about 56% of heterozygous females aged <38 years, in 86% of those aged >38 years and in 100% carriers aged >45 years.

A systematic review of risk factors in Fabry heart disease that included 13 studies for a total of 4185 patients—with a follow-up period of 1.2–10 years—revealed 8.3% of deaths of these, while 75% had cardiovascular causes and 62% were attributable to sudden cardiac death (SCD), a leading cause of cardiovascular mortality in Fabry disease [98]. The mean prevalence of ventricular tachycardia was 15.3%, while age, male gender, LVH, late gadolinium enhancement (LGE) on MRI and Non-Sustained Ventricular Tachycardia (NSVT) were associated with SCD [98].

Niemann et al. [99] demonstrated a clear difference in Fabry cardiomyopathy between males and females, when assessed with cardiac MRI and LGE. Unlike in male patients, loss of myocardial function and the development of fibrosis were not necessarily related to myocardial hypertrophy in female carriers.

Finally, Wilcox et al., analysing the data from the Fabry Registry, a global clinical effort to collect longitudinal data on Fabry disease, reported that of the 1077 females enrolled, 69.4% had symptoms and signs of Fabry disease. The median age at onset of symptoms in females was 13 years, and although about 84% had a positive family history, the diagnosis was delayed up to a mean age of about 30 years. Twenty percent experienced major cerebrovascular, cardiac or renal events at a mean age of 46 years. Among adult females (*n* = 638) in whom glomerular filtration rate (eGFR) was estimated, 62.5% had an eGFR < 90 mL/min/1.73 m^2^ and 19.0% had eGFR <60 mL/min/1.73 m^2^. Proteinuria at 300 mg/day was present in 39.0% and > 1 g/day in 22.2%. The Quality of life (QoL), as measured by the SF-36((R)) survey, was impaired at an older age compared with males, but both genders experience significantly reduced QoL from the third decade of life onward. The authors concluded that Fabry’s carriers have a significant risk for major organ involvement and decreased QoL, and therefore, they should be carefully monitored for a precise estimation of signs and symptoms, as well as adequate therapy [12].

## 3. Skewed X-Chromosome Inactivation

Clinical symptoms in carriers of X-linked diseases depend on the levels of the main protein in the affected tissues. Several mechanisms have been hypothesized such as gene mutation on both alleles [100], loss of one X chromosome as in Turner’s Syndrome [101,102], uniparental disomy [103] or skewed X chromosome inactivation (XCI), with preferential inactivation of the X chromosome carrying the normal allele [92,104,105,106,107,108,109,110].

XCI is an epigenetic mechanism that equalizes the dosage of X-linked genes between sexes through the inactivation of one X chromosome in females. At the end of the process, the females are a mosaic of two cell types expressing either the maternal or paternal X chromosome. Random XCI indicates that about 50% of the cells presents the inactivation of the maternal or paternal X chromosome, while skewed XCI indicates an inactivation of the maternal or paternal X chromosome higher than 50%, usually a ratio of 75:25 or 80:20 between the two chromosomes. The term “extremely skewed XCI” indicates the preferential inactivation of one X chromosome in more than 90% of cells [111] (Figure 1).

Skewed XCI could depend by several factors, such as genetic mechanisms, as the mutations in the X-inactive specific transcript (XIST) gene, involved in familiar cases of skewed XCI [112,113]. Another factor is the plasticity, as cells with high turnover, such as hematopoietic cells, show a higher skewed XCI than cells with lower mitotic activity [114,115]. Skewed XCI in many older females may be related to selection, indeed, a growth or survival advantage conferred by one of the parental X chromosomes [116].

### 3.1. Skewed XCI in the Normal (Healthy) Asymptomatic Females

Previous studies reported a normal distribution of the XCI pattern in the general female population and an extremely skewed XCI in about 5% [116,117,118]. Moreover, XCI correlates with age and type of tissue [119,120]. In particular, the prevalence of skewed XCI is about 16–37% in females over 60 years of age and 49% in centenarians, while it is 14% in females aged ≤ 25 years, and 4.9–14.2% in newborns [118,121,122]. The prevalence of an extremely skewed XCI is about 16–27% in females aged ≥ 60 years and 18% in the centenarians [115,121,122], while it is 7% in females aged ≤ 25 years, and 0.7–2.7% in newborns [118,122,123]. However, a higher percentage of a skewing (about 27%) and an extreme skewing (about 5%) was reported in mothers compared to their newborns, suggesting that hematopoietic cells are affected by age [118]. Considering the type of tissue, a good correlation was reported between blood and epithelial tissue of the same individual [114,115,118,119], as well as between thyroid and muscle, or leucocytes and muscle [108], suggesting that tissues deriving from the same embryogenic layer have the same XCI pattern [105].

### 3.2. Skewed XCI in Carriers of Inherited X-Linked Disorders

Previous studies have shown the correlation between skewed XCI and phenotype in carriers of X-linked diseases, such as Duchenne and Becker muscular dystrophies [104,106,107,108], EDMD1 or myotubular myopathy [124], haemophilia B [125,126], dyskeratosis congenita [127], retinitis pigmentosa [128], Lesch-Nyhan disease, haemophilia A [110,125], Rett-syndrome and others. In fact, skewed XCI, resulted in a higher percentage of mutant cells than normal cells and could lead to clinical symptoms in X-linked disease (Figure 2).

However, despite a large number of studies, the correlation is still debated. It is possible to argue that contradictory results depend on several factors, such as (i) analysis of non-homogeneous groups of carriers and absence of control groups [129,130]; (ii) age of patients, with the analysis of only young patients, who can develop clinical symptoms later in life [131]; (iii) lack of identification of the allele carrying the mutant gene [129,132]; (iv) use of different cut-off for skewed XCI [104,105,108,129,131,132,133]; (v) non-X-linked alleles that can modify the phenotype [134]. Finally, the type of tissue analyzed is of extreme importance, as XCI performed in leukocytes could not reflect the XCI pattern of other cells, which have a different embryogenic origin. [135,136]. The XCI pattern seems not to be inherited, excluding the familiar cases due to XIST mutation, because the mother-daughters did not show the same pattern, as demonstrated in the carriers of dystrophinopathies [106,108], and no concordance is found in the monozygotic female twins [137].

## 4. Meta-Analysis

To assess the possible role of skewed XCI in the clinical manifestations of Fabry carriers, all the studies to date published that examined the XCI in Fabry carriers were considered for the meta-analysis, using the words “Fabry inactivation”, database search engines, PubMed http://www.pubmed.com (accessed on 2 January 2021), Scopus http://www.scopus.com/ (accessed on 2 January 2021) and Cocraine reviews https://www.cochranelibrary.com (accessed on 2 January 2021). We considered as criteria of inclusion research work in English, in which the analysis of XCI was performed in the leukocyte, and the Mainz Severity Score Index (MSSI) and/or cardiac involvement was reported.

Out of 151 identified records, only 8 articles were eligible for the analysis (Figure 3). Of them, only 3 studies reported the MSSI score, and 4 the cardiac involvement and the analysis of XCI pattern (Table 1).

Dobrovolny et al. [138] analyzed the pattern of XCI in 38 female carriers, of which only 11 (28.9%) exhibited skewed XCI considering the cut-off of 75:25. Moreover, 24 females showed a MMSI < 20 (63.1%), and 14 female MSSI > 20 (36.8%), but of these only 4 (28.6%) were skewed. There was a correlation between age onset and skewed XCI. No data on cardiomyopathy in female with random XCI were reported, so it was not included in the meta-analysis on the correlation between cardiac involvement and skewed XCI.

Maier et al. [139] analyzed the XCI pattern in 28 symptomatic carriers, considering a different cut-off, in particular the ratio between the two X chromosome 65:35 as moderate skewing, and a ratio of greater than 80:20 as extremely skewed. Thirteen (46%) females showed random XCI, ten (36%) moderate skewing, and five (18%) highly skewed XCI. They also analyzed 56 healthy females, reporting respectively random XCI in 29 (52%), moderate skewing in 16 female (29%), and highly skewed patterns of XCI in 11 subjects (20%) without any significant difference.

If we consider the ratio of 75:25, taking the data by the table reported in this study, out of 28 symptomatic carriers, ten female (35.7%) presented skewed XCI, of which 5 (50%) presented a MSSI > 20. While 13 female (46.4%) with random XCI showed a MSSI > 20. Out of 16 females with skewed XCI, 6 (37.5%) showed cardiovascular involvement. The authors conclude that skewed XCI is not correlated with phenotype.

Echevarria et al. [140] presented the data on a cohort of 56 female carriers, of which 52 were informative for XCI analysis. Ten female (19.2%) showed a preferential inactivation of the X chromosome carrying the normal allele, of them 7presented a MSSI > 20. Out of 10 skewed females, 7 (70%) showed increase LVM. The authors also analyzed different types of tissue (blood, skin, buccal smears, urine), demonstrated no correlation between XCI pattern in the blood and the others tissue. Moreover, they showed a correlation between age and severity of phenotype, and between skewed XCI and age onset.

Morrone et al. [141] and Rossanti et al. [142] analyzed a limited number of carriers (9 and 4 subjects, respectively). They did not presented data on MSSI in Fabry carriers, while the cardiomyopathy was reported in 5/13 carriers, but only 1 showed a skewed XCI pattern in the leukocytes.

The Fabry carriers from these studies were divided into two groups: symptomatic and asymptomatic, based on the Mainz Severity Score Index (MSSI) and/or the presence of cardiac involvement. Considering the severity of symptoms according to the MSSI score, symptomatic carriers were further subdivided in two subgroups, mild (MSSI < 20) vs. moderate-severe (MSSI > 20).

The meta-analysis was performed with the Pro-Meta 3. The statistical heterogeneity among the studies was assessed with the I-squared test (I^2^), where I^2^ value ≥ 75% represents a large heterogeneity between studies.

In this model, only one study included in the analysis showed a statistically significant association between skewed XCI and disease severity. The weight given for Dobrovolny et al., Echeivarra et al., Maier et al. was 34.3, 33.7 and 31.9%, respectively. The results of the meta-analysis are shown in the forest plot (Figure 4).

No preferential inactivation of the X chromosome carrying the normal allele was confirmed in mild compared to moderate-severe symptomatic carriers [OR 0.78 (95% CI 0.18–3.46), *p* = 0.74].

When looking at skewed XCI and cardiac involvement, none of the studies included in the analysis showed a statistically significant association between skewed XCI and phenotype. The weight given for Echeivarra et al., Maier et al., Morrone et al., and Rossanti et al. was 40.59, 44.63, 5.97 and 8.82%, respectively. The results of the meta-analysis are shown in the forest plot (Figure 5). Also in this case, no preferential inactivation of the X chromosome carrying the normal allele was confirmed in symptomatic compared to asymptomatic carriers [OR 0.6 (95% CI 0.221–1.7), *p* = 0.332].

## 5. Discussion

Fabry disease involves several tissues, leading to a complex phenotype including skin lesions, cardiac, renal and nervous system complications. The key of pathogenesis is the accumulation of Gb3 determined by the absent or decreased levels of α-gal in different types of cells, in particular in lysosomes but also in the cytosol. The excessive store of Gb3 increases the level of Lyso-Gb3, and triggers the inflammatory cascade, that in turn causes ischemia, fibrosis and apoptosis.

As in other X-linked disorders, the presence of cell mosaicism is considered an advantage for female carriers, because random XCI usually results in the presence of wild-type alleles in at least 50% of the cells, allowing female carriers not to show clinical symptoms. Furthermore, the interaction between the cells expressing the wild-type or mutant X chromosome determines a metabolic cooperation which leads to the correction, at least in part, of the defect in the mutant cells [143,144]. This appears to occur in X-linked lysosomal disorders, such as Hunter and Fabry diseases, in which the normal enzyme is transferred to a mutant cell by mannose-6-phosphate–mediated endocytosis [116]. Like the carriers of muscular dystrophies (MD), such as Duchenne/Becker (DMD/BMD) or Emery-Dreifuss (EDMD), Fabry female carriers can also exhibit clinical symptoms, and in particular cardiac involvement, either as cardiomyopathy or conduction system defects. The preferential inactivation of the X chromosome carrying the normal allele has suggested to explain the onset of symptoms in female carriers.

In fact, a significant correlation was found between cardiomyopathy and skewed XCI assessed in leukocytes in BMD/DMD carriers, suggesting that the analysis of XCI in this tissue can be useful to predict the phenotype [107]. This correlation has also been explored in recent studies on carriers of Fabry disease. Echevarria et al. [140], analyzing the human androgen receptor gene in females with Fabry disease, showed a positive correlation between a skewed XCI and disease severity. Dobrovolny et al. [138] found a statistically significant difference between the severity score values of heterozygotes with random and non-random X-chromosome inactivation, suggesting that X-inactivation is a major factor determining the severity of clinical involvement in Fabry heterozygotes, and that its status could serve as an important indicator for an early, presymptomatic treatment of heterozygotes. Conversely, other studies [139,142] reported that X inactivation patterns in symptomatic females, heterozygous for Fabry disease, did not differ from those of female controls of the same age.

The present meta-analysis failed to observe any correlation between clinical symptoms, including cardiac involvement and skewed XCI.

This apparent discrepancy can be linked to several factors: first, it cannot be excluded those carriers with skewed XCI (≥75:25) and no symptoms at the time of evaluation, will develop them later with age. In fact, the number of symptomatic females analysed in the study of Dobrovolny et al. [138] or Maier et al. [139] are younger than 40 y. Second, the type of cardiac involvement, cardiomyopathy or conduction system defects, was not specified; third, it cannot ruled out that the XCI pattern analyzed in leukocytes is not always useful to predict carrier phenotypes or disease progression. XCI is usually assessed in leukocytes from peripheral blood because it is the most common source of DNA supply, but, as suggested for EDMD1 carriers [145], the discordance between cardiac phenotype (conduction system defects) and degree of the XCI can be explained as likely depending by the different embryological origin of the tissues analyzed (constituents of the conduction cardiac tissue and blood).

The same explanation (hypothesis) could apply to Fabry disease, where the pathogenesis of cardiac involvement can be related both to the involvement of cardio-myocytes leading to cardiomyopathy, and to the involvement of conduction system leading to arrhythmias [146]. The different embryological origin of the tissues analyzed could also explain the absence of correlation observed between skewed XCI and the MSSI score. MSSI is in fact focused on different target organs, such as the neurological system, heart and kidney, which all have an embryological origin different from blood.

In conclusion, we hypothesize that the analysis of XCI in leukocytes is not always useful for predicting the phenotype in Fabry carriers, and suggest extending to other tissues the study of XCI to better evaluate the correlation between skewed XCI and the severity of symptoms.

## Figures and Tables

**Figure 1 ijms-22-07663-f001:**
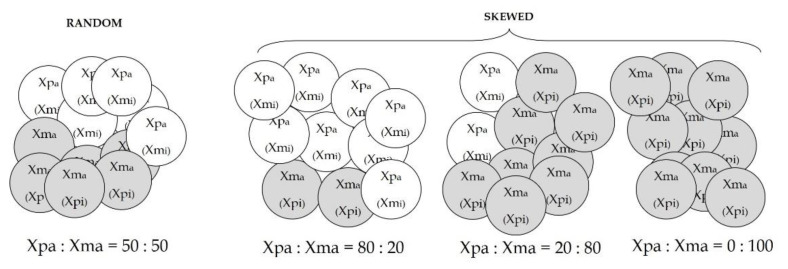
Pattern of XCI in healthy females. Each tissue of females is a mosaic of cells, including cells (white circle) in which the paternal X chromosome is activated (Xpa) and the maternal X chromosome is inactivated (Xmi) and cells (grey circle) in which the maternal X chromosome is activated (Xma) and the paternal X chromosome is inactivated (Xpi).

**Figure 2 ijms-22-07663-f002:**
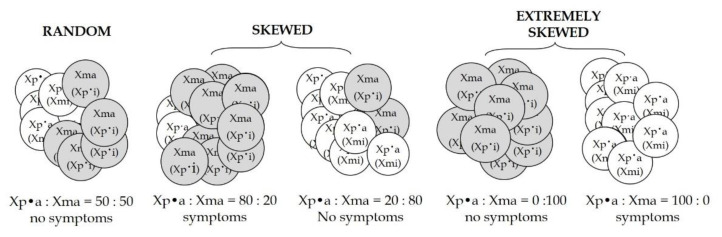
Example of cellular mosaicism in a female carrier of X-linked disease. The figure represents an example of (a) a random XCI pattern with the same percentage of cells expressing the paternal (white) mutated X chromosome (Xp•) and the maternal (grey) X chromosome (Xm) (on the left); (b) a skewed XCI pattern with a higher percentage of cells expressing the maternal X chromosome that determines the absence of symptoms or a skewed XCI pattern with a higher percentage of cells expressing the Xp• that determines the presence of symptoms (in the center); (c) an extremely skewed XCI with a higher percentage of cells expressing the maternal X chromosome that determines the absence of symptoms or an extremely skewed XCI pattern with a higher percentage of cells expressing the Xp• that determines the presence of symptoms (on the right). a = activated; i = inactivated.

**Figure 3 ijms-22-07663-f003:**
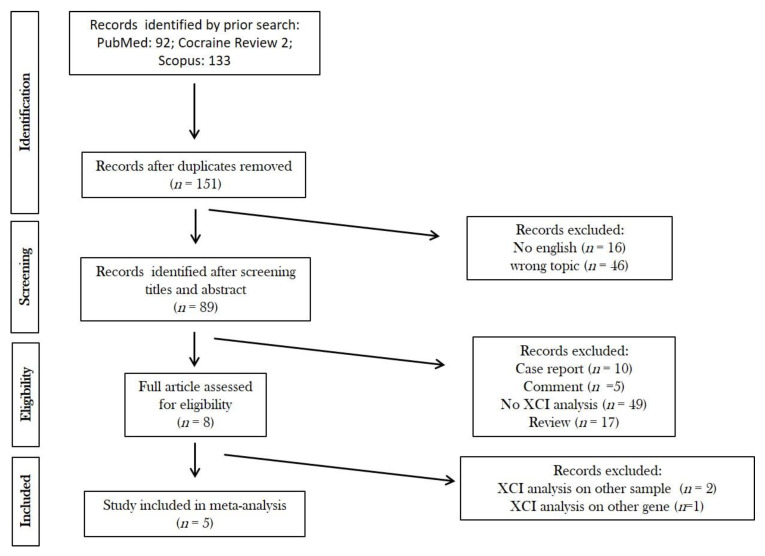
Flow chart.

**Figure 4 ijms-22-07663-f004:**
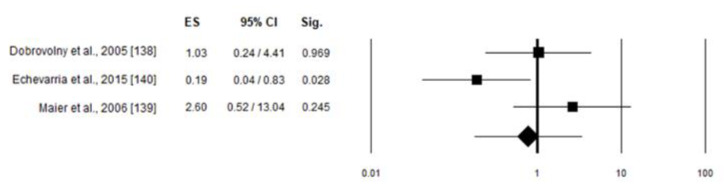
Forest plots for skewed versus random XCI in mild and moderate-severe symptomatic Fabry carriers. Heterogeneity I^2^ = 65.43%, *p* = 0.055.

**Figure 5 ijms-22-07663-f005:**
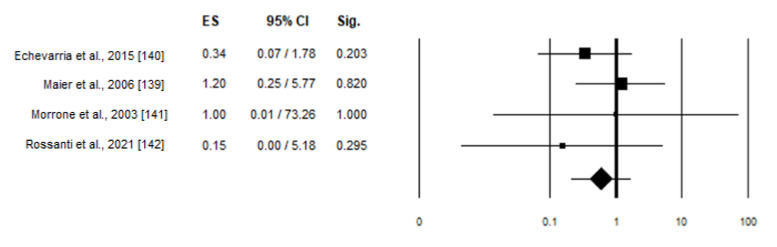
Forest plots for skewed versus random XCI in symptomatic Fabry carriers at cardiac level. Heterogeneity I^2^ = 0.0%, *p* = 0.609.

**Table 1 ijms-22-07663-t001:** Results of studies on the XCI in Fabry carriers.

Articles	Age		Skewed XCI			
		Tissue Analyzed	Mild MSSI Score (Total Subjects)	Moderate-Severe MSSI Score (Total Subjects)	Cardiac Involvement (Total Subjects)	No Cardiac Involvement (Total Subjects)
Dobrovolny et al., 2005	Young/Adult	L, U, SE	7 (24)	4 (14)	n.d.	n.d.
Maier et al., 2006	Young/Adult	L	5 (10)	5 (18)	6 (16)	4 (12)
Echeivarra et al., 2015	Young/Adult	L, U, SE, skin	3 (35)	7 (21)	7 (41)	3 (8)
Morrone et. al., 2003	Young/Adult	L	n.d.	n.d.	2 (0)	2 (4)
Rossanti et al., 2021	Adult	L,	n.d.	n.d.	0 (5)	1 (2)

n.d. = not determined; L = peripheral blood leukocytes; SE = salivary epithelia; U = urinary sediment cells; skewed XCI is referred to preferential inactivation of wild X chromosome with a ratio of 75:25.

## Data Availability

Not Applicable.

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
