# Peer review of "X Chromosome Inactivation in Carriers of Fabry Disease: Review and Meta-Analysis"

_ijms, 2021, doi:10.3390/ijms22147663_

Round 1
Reviewer 1 Report
In the review “X-chromosome inactivation in carriers of Fabry disease: review and meta-analysis” the authors discuss the issue of female carriers in Fabry disease and present a meta-analysis about the correlation of female Fabry’s phenotypes and skewed X-chromosome inactivation.
Overall, the paper is flowing and well-written. The introduction provides a good overview of the topic, even if some references need to be included. The methods are clearly presented, and the results discussed in a proper way. The conclusions are supported by the results obtained.
Even so, some issues need to be addressed, particularly regarding the methodology used for the meta-analysis.
Major issues
Introduction
Paragraph 1.3
The authors briefly mention the problem of genotype-phenotype in Fabry disease. This represents a major issue in the field. Even if it is justifiable that the argument is not discussed into details, a few references should be added, to give the readers a more complete overview of the literature. Some examples of papers on the topic are here suggested:
Ries M, Gal A. Genotype–phenotype correlation in Fabry disease. In: Mehta A, Beck M, Sunder-Plassmann G, editors. Fabry Disease: Perspectives from 5 Years of FOS. Oxford: Oxford PharmaGenesis; 2006. Chapter 34. PMID: 21290681.
Pan X, Ouyang Y, Wang Z, Ren H, Shen P, Wang W, Xu Y, Ni L, Yu X, Chen X, Zhang W, Yang L, Li X, Xu J, Chen N. Genotype: A Crucial but Not Unique Factor Affecting the Clinical Phenotypes in Fabry Disease. PLoS One. 2016 Aug 25;11(8):e0161330. doi: 10.1371/journal.pone.0161330. PMID: 27560961; PMCID: PMC4999276.
Koca S, Tümer L, Okur İ, Erten Y, Bakkaloğlu S, Biberoğlu G, Kasapkara Ç, Küçükçongar A, Dalgıç B, Oktar SÖ, Öner Y, Atalay T, Cemri M, Çiftçi B, Topçu B, Hasanoğlu A, Ezgü F. High incidence of co-existing factors significantly modifying the phenotype in patients with Fabry disease. Gene. 2019 Mar 1;687:280-288. doi: 10.1016/j.gene.2018.11.054. Epub 2018 Nov 20. PMID: 30468909.
Mignani R, Moschella M, Cenacchi G, Donati I, Flachi M, Grimaldi D, Cerretani D, Giovanni P, Montevecchi M, Rigotti A, Ravasio A. Different renal phenotypes in related adult males with Fabry disease with the same classic genotype. Mol Genet Genomic Med. 2017 May 8;5(4):438-442. doi: 10.1002/mgg3.292. PMID: 28717668; PMCID: PMC5511792.
Lukas J, Giese AK, Markoff A, Grittner U, Kolodny E, Mascher H, Lackner KJ, Meyer W, Wree P, Saviouk V, Rolfs A. Functional characterisation of alpha-galactosidase a mutations as a basis for a new classification system in fabry disease. PLoS Genet. 2013;9(8):e1003632. doi:
Paragraph 1.4
The authors briefly mention the treatment for Fabry disease.
As for the paragraph 1.3, given the conciseness of the paragraph it could be useful to provide literature references to the interested readers. In fact, on the side of ERT, use of the approved pharmacological chaperone DGJ is a general medical practice. The stratification of patients and the prediction of their responsiveness to PC therapy is a hot topic in the field of Fabry disease. The authors should mention the pharmacological chaperone therapy.
Some references are suggested:
Felis A, Whitlow M, Kraus A, Warnock DG, Wallace E. Current and Investigational Therapeutics for Fabry Disease. Kidney Int Rep. 2019 Dec 6;5(4):407-413. doi: 10.1016/j.ekir.2019.11.013. PMID: 32274449; PMCID: PMC7136345.
Simonetta I, Tuttolomondo A, Daidone M, Miceli S, Pinto A. Treatment of Anderson-Fabry Disease. Curr Pharm Des. 2020;26(40):5089-5099. doi: 10.2174/1381612826666200317142412. PMID: 32183665.
Citro V, Cammisa M, Liguori L, Cimmaruta C, Lukas J, Cubellis MV, Andreotti G. The Large Phenotypic Spectrum of Fabry Disease Requires Graduated Diagnosis and Personalized Therapy: A Meta-Analysis Can Help to Differentiate Missense Mutations. Int J Mol Sci. 2016 Dec 1;17(12):2010. doi: 10.3390/ijms17122010. PMID: 27916943; PMCID: PMC5187810.
Cammisa, M., Correra, A., Andreotti, G., & Cubellis, M. V. (2013). Fabry_CEP: a tool to identify Fabry mutations responsive to pharmacological chaperones. Orphanet journal of rare diseases, 8(1), 1-3. PMID: 23883437 PMCID: PMC3729670
Liguori, L., Monticelli, M., Allocca, M., Hay Mele, B., Lukas, J., Cubellis, M. V., & Andreotti, G. (2020). Pharmacological chaperones: A therapeutic approach for diseases caused by destabilizing missense mutations. International journal of molecular sciences, 21(2), 489PMID: 31940970 PMCID: PMC7014102
Porto C, Pisani A, Rosa M, Acampora E, Avolio V, Tuzzi MR, Visciano B, Gagliardo C, Materazzi S, la Marca G, Andria G, Parenti G. Synergy between the pharmacological chaperone 1-deoxygalactonojirimycin and the human recombinant alpha-galactosidase A in cultured fibroblasts from patients with Fabry disease. J Inherit Metab Dis. 2012 May;35(3):513-20. doi: 10.1007/s10545-011-9424-3. Epub 2011 Dec 21. PMID: 22187137.
Hughes DA, Nicholls K, Shankar SP, Sunder-Plassmann G, Koeller D, Nedd K, Vockley G, Hamazaki T, Lachmann R, Ohashi T, Olivotto I, Sakai N, Deegan P, Dimmock D, Eyskens F, Germain DP, Goker-Alpan O, Hachulla E, Jovanovic A, Lourenco CM, Narita I, Thomas M, Wilcox WR, Bichet DG, Schiffmann R, Ludington E, Viereck C, Kirk J, Yu J, Johnson F, Boudes P, Benjamin ER, Lockhart DJ, Barlow C, Skuban N, Castelli JP, Barth J, Feldt-Rasmussen U. Oral pharmacological chaperone migalastat compared with enzyme replacement therapy in Fabry disease: 18-month results from the randomised phase III ATTRACT study. J Med Genet. 2017 Apr;54(4):288-296. doi: 10.1136/jmedgenet-2016-104178. Epub 2016 Nov 10. Erratum in: J Med Genet. 2018 Apr 16;: PMID: 27834756; PMCID: PMC5502308.
Meta-analysis – paragraph 4
In the text (lines 270-274) the authors declare the search was performed using PubMed and Scopus, but in the flow chart in Figure 3, the Cochrane Library is mentioned. Also, in the text (line 275) the state that “8 articles were eligible for analysis”, but in the flow-chart in Figure 3 there are 7 articles assessed for eligibility. These points should be clearer.
In my opinion, the authors should justify why they chose to use only one combination of words (“Fabry inactivation”) to perform the research. They should try whether different words’ combinations provide results that were not included in the present form of the literature search.
Minor issues
The authors refer to alpha-galactosidase as GLA. Actually, GLA is the gene name (and it should be italics formatted), but α-gal (or AGAL) is the protein name. The nomenclature is not homogenous and should be revised through the article.
Lane 75: some words are probably missing
Lane 79: redundant comma
Lane 241: useless link to be removed
Lanes 292-293: syntax needs to be checked, something is probably missing
Author Response
Reply to Reviewers
We are very grateful to the reviewers for their constructive comments and suggestions, which allowed us to improve the manuscript. Below are the answers point by point.
Reviewer 1
In the review “X-chromosome inactivation in carriers of Fabry disease: review and meta-analysis” the authors discuss the issue of female carriers in Fabry disease and present a meta-analysis about the correlation of female Fabry’s phenotypes and skewed X-chromosome inactivation.
Overall, the paper is flowing and well-written. The introduction provides a good overview of the topic, even if some references need to be included. The methods are clearly presented, and the results discussed in a proper way. The conclusions are supported by the results obtained.
Even so, some issues need to be addressed, particularly regarding the methodology used for the meta-analysis.
Major issues
Introduction
Paragraph 1.3
The authors briefly mention the problem of genotype-phenotype in Fabry disease. This represents a major issue in the field. Even if it is justifiable that the argument is not discussed into details, a few references should be added, to give the readers a more complete overview of the literature. Some examples of papers on the topic are here suggested:
Ries M, Gal A. Genotype–phenotype correlation in Fabry disease. In: Mehta A, Beck M, Sunder-Plassmann G, editors. Fabry Disease: Perspectives from 5 Years of FOS. Oxford: Oxford PharmaGenesis; 2006. Chapter 34. PMID: 21290681.
Pan X, Ouyang Y, Wang Z, Ren H, Shen P, Wang W, Xu Y, Ni L, Yu X, Chen X, Zhang W, Yang L, Li X, Xu J, Chen N. Genotype: A Crucial but Not Unique Factor Affecting the Clinical Phenotypes in Fabry Disease. PLoS One. 2016 Aug 25;11(8):e0161330. doi: 10.1371/journal.pone.0161330. PMID: 27560961; PMCID: PMC4999276.
Koca S, Tümer L, Okur İ, Erten Y, Bakkaloğlu S, Biberoğlu G, Kasapkara Ç, Küçükçongar A, Dalgıç B, Oktar SÖ, Öner Y, Atalay T, Cemri M, Çiftçi B, Topçu B, Hasanoğlu A, Ezgü F. High incidence of co-existing factors significantly modifying the phenotype in patients with Fabry disease. Gene. 2019 Mar 1;687:280-288. doi: 10.1016/j.gene.2018.11.054. Epub 2018 Nov 20. PMID: 30468909.
Mignani R, Moschella M, Cenacchi G, Donati I, Flachi M, Grimaldi D, Cerretani D, Giovanni P, Montevecchi M, Rigotti A, Ravasio A. Different renal phenotypes in related adult males with Fabry disease with the same classic genotype. Mol Genet Genomic Med. 2017 May 8;5(4):438-442. doi: 10.1002/mgg3.292. PMID: 28717668; PMCID: PMC5511792.
Lukas J, Giese AK, Markoff A, Grittner U, Kolodny E, Mascher H, Lackner KJ, Meyer W, Wree P, Saviouk V, Rolfs A. Functional characterisation of alpha-galactosidase a mutations as a basis for a new classification system in fabry disease. PLoS Genet. 2013;9(8):e1003632. doi:
Response: We thank the reviewer for the suggestion; we have now expanded the paragraph and added the references suggested (n. 49-53).
Paragraph 1.4
The authors briefly mention the treatment for Fabry disease.
As for the paragraph 1.3, given the conciseness of the paragraph it could be useful to provide literature references to the interested readers. In fact, on the side of ERT, use of the approved pharmacological chaperone DGJ is a general medical practice. The stratification of patients and the prediction of their responsiveness to PC therapy is a hot topic in the field of Fabry disease. The authors should mention the pharmacological chaperone therapy.
Some references are suggested:
Felis A, Whitlow M, Kraus A, Warnock DG, Wallace E. Current and Investigational Therapeutics for Fabry Disease. Kidney Int Rep. 2019 Dec 6;5(4):407-413. doi: 10.1016/j.ekir.2019.11.013. PMID: 32274449; PMCID: PMC7136345.
Simonetta I, Tuttolomondo A, Daidone M, Miceli S, Pinto A. Treatment of Anderson-Fabry Disease. Curr Pharm Des. 2020;26(40):5089-5099. doi: 10.2174/1381612826666200317142412. PMID: 32183665.
Citro V, Cammisa M, Liguori L, Cimmaruta C, Lukas J, Cubellis MV, Andreotti G. The Large Phenotypic Spectrum of Fabry Disease Requires Graduated Diagnosis and Personalized Therapy: A Meta-Analysis Can Help to Differentiate Missense Mutations. Int J Mol Sci. 2016 Dec 1;17(12):2010. doi: 10.3390/ijms17122010. PMID: 27916943; PMCID: PMC5187810.
Cammisa, M., Correra, A., Andreotti, G., &Cubellis, M. V. (2013). Fabry_CEP: a tool to identify Fabry mutations responsive to pharmacological chaperones. Orphanet journal of rare diseases, 8(1), 1-3. PMID: 23883437 PMCID: PMC3729670
Liguori, L., Monticelli, M., Allocca, M., Hay Mele, B., Lukas, J., Cubellis, M. V., & Andreotti, G. (2020). Pharmacological chaperones: A therapeutic approach for diseases caused by destabilizing missense mutations. International journal of molecular sciences, 21(2), 489PMID: 31940970 PMCID: PMC7014102
Porto C, Pisani A, Rosa M, Acampora E, Avolio V, Tuzzi MR, Visciano B, Gagliardo C, Materazzi S, la Marca G, Andria G, Parenti G. Synergy between the pharmacological chaperone 1-deoxygalactonojirimycin and the human recombinant alpha-galactosidase A in cultured fibroblasts from patients with Fabry disease. J Inherit Metab Dis. 2012 May;35(3):513-20. doi: 10.1007/s10545-011-9424-3. Epub 2011 Dec 21. PMID: 22187137.
Hughes DA, Nicholls K, Shankar SP, Sunder-Plassmann G, Koeller D, Nedd K, Vockley G, Hamazaki T, Lachmann R, Ohashi T, Olivotto I, Sakai N, Deegan P, Dimmock D, Eyskens F, Germain DP, Goker-Alpan O, Hachulla E, Jovanovic A, Lourenco CM, Narita I, Thomas M, Wilcox WR, Bichet DG, Schiffmann R, Ludington E, Viereck C, Kirk J, Yu J, Johnson F, Boudes P, Benjamin ER, Lockhart DJ, Barlow C, Skuban N, Castelli JP, Barth J, Feldt-Rasmussen U. Oral pharmacological chaperone migalastat compared with enzyme replacement therapy in Fabry disease: 18-month results from the randomised phase III ATTRACT study. J Med Genet. 2017 Apr;54(4):288-296. doi: 10.1136/jmedgenet-2016-104178. Epub 2016 Nov 10. Erratum in: J Med Genet. 2018 Apr 16;: PMID: 27834756; PMCID: PMC5502308.
Response: We thank the reviewer for the suggestion; we have now expanded the paragraph and added some of the references suggested (n. 62-63)
Meta-analysis – paragraph 4
In the text (lines 270-274) the authors declare the search was performed using PubMed and Scopus, but in the flow chart in Figure 3, the Cochrane Library is mentioned. Also, in the text (line 275) the state that “8 articles were eligible for analysis”, but in the flow-chart in Figure 3 there are 7 articles assessed for eligibility. These points should be clearer.
Response: We have now correcedt the error in the text.
In my opinion, the authors should justify why they chose to use only one combination of words (“Fabry inactivation”) to perform the research. They should try whether different words’ combinations provide results that were not included in the present form of the literature search.
Response: We agree with this point. We have also used a combination as “Fabry X chromosome inactivation”, but more studies were found using a wider research. However, we have now added two studies in the analysis.
Minor issues
The authors refer to alpha-galactosidase as GLA. Actually, GLA is the gene name (and it should be italics formatted), but α-gal (or AGAL) is the protein name. The nomenclature is not homogenous and should be revised through the article.
Response: We now indicate the gene and the protein, according to your suggestions, throughout the manuscript.
Lane 75: some words are probably missing
Response: Thanks for this note. We have now changed the text
Lane 79: redundant comma
Response: We eliminated the redundant comma
Lane 241: useless link to be removed
Response: We did it
Lanes 292-293: syntax needs to be checked, something is probably missing
Response: Thanks for this note. We have now corrected the text.

Reviewer 2 Report
The article “X Chromosome Inactivation in Carriers of Fabry Disease: Re-2 view and Meta-Analysis” by Emanuela Viggiano and Luisa Politano approaches on the subject of correlation between skewed X chromosome inactivation (XCI) and symptom severity in female carriers of X-linked Fabry disease. I think the idea of combining the review with a meta-analysis is very good.
The article is overall well written and deals with an interesting subject which, from my perspective, has not yet been the subject of much controversial written discussion outside of expert congresses. Therefore, I totally dig the approach, but have some complaints about the script that I would subject to revision. Point-by-point in chronological order (not sorted by importance):
- GLA gene should be written italic throughout the script. In order to better distinguish protein from gene, here some suggestions as abbreviations for the protein: AGAL, α-Gal, α-GalA
- Although not the direct focus of the paper, the assessment of clinical symptoms (1.2 Clinical Presentation) is not insignificant when considering female carriers, so this should also be done accurately. This part is quite superficial and, in my view, not entirely accurate. Example: Hyperhidrosis is a rare sign of FD; the common sign is hypohydrosis.
- The sentence: “Cardiac involvement in Fabry disease is frequent and is more severe in hemizygotes.” (section: 1.2.1. Cardiac manifestations). Does this refer to a comparison between hemizygote males and heterozygous females? I think this indicates that females do less frequently suffer from cardiac symptoms in general? This does need seem to be the case after having read the whole article. This should be clarified and a reference should be provided.
- Line 100: studies (instead of studied).
- I find this section (1.2.1. Cardiac manifestations) a bit confused. The role of Gb3 and lyso-Gb3 on the pathophysiology in FD is generally not clear. Why are cardiac symptoms used as an example here? I found the role of Gb3 on the heart more conclusively described in Seydelman et al. (Best Practice & Research Clinical Endocrinology & Metabolism
- Volume 29, Issue 2, March 2015, Pages 195-204). Example: “It has been shown that the storage of Gb3 induces an excessive production of reactive oxygen species in cultured vascular endothelial cells thereby increasing oxidative stress. Gb3 also up-regulates the expression of adherence molecules in vascular endothelium [16]. Other data indicate that Gb3 may cause the release of pro-inflammatory cytokines, especially dendritic cells and monocytes [4]. Thus, it can be hypothesized that Gb3 storage triggers a cascade of pathophysiological processes leading to a structural cellular change, tissue defects, and – over time – to organ failure.” This (≈line 92-122) should be re-structured. The rationale why this aspect was singled out and is important for the rest of the article should also be mentioned.
- The section on Genotype/Phenotype is very short. Lots of articles approached this subject from the biomarker side (Smid et al, J Med Genet. 2015 Apr;52(4):262-8.) to the enzyme activity side (Lukas et al, PLoS Genet. 2013;9(8):e1003632.).
- Under 1.4 Treatment the treatment options should be mentioned, at least briefly. ERTs from several providers are available and since 2016 the Chaperone Galafold is approaved by FDA and EMA. I think it is worth a mentioning since it is the first Pharmacochaperone approved for an LSD.
- “…but both genders experience significantly reduced QoL from the third decade of life onward.” Gender identity does not refer to the biological sex. But here the biological sex should be used.
- Line 191: “…loss of one X chromosome [66,67],…” shouldn´t it read “partial loss” in order to avoid misunderstandings?
- I do not agree with the definition of random and skewed X-inactivation. I am of the opinion that up to a degree of 70:30 one can assume random X-inactivation, only at more extreme ratios one can speak of skewed. However, I am not sure if there is a precise definition. This should be defined precisely. A ratio of exactly 50:50 should be the most frequent with an assumed normal distribution, but all possible intermediate states are conceivable.
- The legend for Figure 1 describes the gray and white cells in a wrong way. It should state: “(gray – Xma)” instead of “(gray – Xmi)”.
- The sentence “Cellular mosaicism is an advantage for females in X-linked diseases, because XCI is usually random and results in the presence of wild-type alleles in at least 50% of cells. In this way, wild-type gene expression allows female carriers to show no clinical symptoms.” Is provided a bit early. If I am not mistaken, this should be the result of the meta-analysis, I guess. Furthermore, I think it is a popular theory and a very good evidence-based explanation for many findings in other X-linked diseases as well. Some relevant references should be brought here to underline the significance of the topic. And the reason why 50% "activity" of a given gene is sufficient for freedom of symptoms should also be addressed, namely the assumed redundancy of the system.
- The section 3.2. Skewed XCI in asymptomatic females deals with whole population data, not with patients with diagnosed diseases or carriers whatsoever, correct? I think it should be named “Skewed inactivation in the normal (or healthy) population”. Asymptomatic is not healthy, It refers to a more specific cohort than just randomized “whole population data”.
- Lines 225-27: “…in fact familiar cases of skewed XCI have been reported, leading to severe clinical symptoms or discordant phenotype.” This is in my opinion contradictory to the sentence in lines 266/67: “The XCI pattern seems not to be inherited, because the mother-daughters did not show the same pattern in the carriers of dystrophinopathies…” For the latter statement two references are provided so I believe you can find evidence for both statements. As it is presented now, I find it contradictory.
- I have some concerns with the decision tree for literature selection for the meta-analysis. 3 articles are very few. Why were only studies with MSSI included? Why were case reports excluded? There are sometimes very interesting conclusions of the treating physicians that could be controversial. I only had a quick look at the literature, but Morrone et al (J Med Genet. 2003 Aug;40(8):e103.) describe 4 female patients, two are asymptomatic (random X-inactivation) and 2 manifest disease symptoms (skewed in favour of mutant chromosome). These are clear indications that this plays an important role. Of course, no clear statistics can then be superimposed on the data, but statistical significance is not everything, sometimes medical relevance is nevertheless given. As in the example of the inheritance of skewedness. It may not be the rule, but it can happen. However, it can be lost in the totality.
- The original studies (Dobrovolny et al., 2005; Maier et al., 2006; Echeivarra et al., 2015) should be better introduced. Table 1 is helpful, of course, but it would be important to know what limit was used to assess skewedness (70:30, 80:20) in the individual studies and were these adopted or did they have to be adjusted for the meta-analysis? Or were the actual values used instead of a binary yes/no?
- One important question for me to fully understand the key message here: Do the original 3 studies individually claim that skewedness is significant for the phenotype and you falsified that? The authors of the original studies should also have come to the conclusion that leukocytes are unsuitable for an analysis of XCI?

Author Response
Reply to Reviewers
We are very grateful to the reviewers for their constructive comments and suggestions, which allowed us to improve the manuscript. Below are the answers point by point.
Reviewer 2
The article “X Chromosome Inactivation in Carriers of Fabry Disease: Review and Meta-Analysis” by Emanuela Viggiano and Luisa Politano approaches on the subject of correlation between skewed X chromosome inactivation (XCI) and symptom severity in female carriers of X-linked Fabry disease. I think the idea of combining the review with a meta-analysis is very good.
The article is overall well written and deals with an interesting subject which, from my perspective, has not yet been the subject of much controversial written discussion outside of expert congresses. Therefore, I totally dig the approach, but have some complaints about the script that I would subject to revision. Point-by-point in chronological order (not sorted by importance):
- GLA gene should be written italic throughout the script. In order to better distinguish protein from gene, here some suggestions as abbreviations for the protein: AGAL, α-Gal, α-GalA
Response: We thank the reviewer; We now indicate the gene and the protein, according to your suggestions throughout the manuscript
- Although not the direct focus of the paper, the assessment of clinical symptoms (1.2 Clinical Presentation) is not insignificant when considering female carriers, so this should also be done accurately. This part is quite superficial and, in my view, not entirely accurate. Example: Hyperhidrosis is a rare sign of FD; the common sign is hypohydrosis.
Response: We thanks the reviewer for the suggestion. We have now corrected the error in the text, and implemented the paragraph.
- The sentence: “Cardiac involvement in Fabry disease is frequent and is more severe in hemizygotes.” (section: 1.2.1. Cardiac manifestations). Does this refer to a comparison between hemizygote males and heterozygous females? I think this indicates that females do less frequently suffer from cardiac symptoms in general? This does need seem to be the case after having read the whole article. This should be clarified and a reference should be provided.
Response: We have now clarified the concept, according to your suggestion and added the reference (n. 40,41,42)
- Line 100: studies (instead of studied).
Response: We have corrected the error in the text.
- I find this section (1.2.1. Cardiac manifestations) a bit confused. The role of Gb3 and lyso-Gb3 on the pathophysiology in FD is generally not clear. Why are cardiac symptoms used as an example here? I found the role of Gb3 on the heart more conclusively described in Seydelman et al. (Best Practice & Research Clinical Endocrinology & Metabolism Volume 29, Issue 2, March 2015, Pages 195-204). Example: “It has been shown that the storage of Gb3 induces an excessive production of reactive oxygen species in cultured vascular endothelial cells thereby increasing oxidative stress. Gb3 also up-regulates the expression of adherence molecules in vascular endothelium [16]. Other data indicate that Gb3 may cause the release of pro-inflammatory cytokines, especially dendritic cells and monocytes [4]. Thus, it can be hypothesized that Gb3 storage triggers a cascade of pathophysiological processes leading to a structural cellular change, tissue defects, and – over time – to organ failure.” This (≈line 92-122) should be re-structured. The rationale why this aspect was singled out and is important for the rest of the article should also be mentioned.
Response: To avoid confusion, we have now inserted a new paragraph on pathophisiology and modified the lines 92-122, as suggested
- The section on Genotype/Phenotype is very short. Lots of articles approached this subject from the biomarker side (Smid et al, J Med Genet. 2015 Apr;52(4):262-8.) to the enzyme activity side (Lukas et al, PLoS Genet. 2013;9(8):e1003632.).
Response: We thank the reviewer for the suggestion, we wanted focalize the review on XCI in Fabry carriers. However, we have expanded the section and added new references.
- Under 1.4 Treatment the treatment options should be mentioned, at least briefly. ERTs from several providers are available and since 2016 the Chaperone Galafold is approaved by FDA and EMA. I think it is worth a mentioning since it is the first Pharmacochaperone approved for an LSD.
Response: We thank the reviewer for the suggestion; we have now expanded the paragraph and added the references suggested.
- “…but both genders experience significantly reduced QoL from the third decade of life onward.” Gender identity does not refer to the biological sex. But here the biological sex should be used.
Response: we used the term “sexes” instead of “genders”.
- Line 191: “…loss of one X chromosome [66,67],…” shouldn´t it read “partial loss” in order to avoid misunderstandings?
Response: we have now explained better this concept.
- I do not agree with the definition of random and skewed X-inactivation. I am of the opinion that up to a degree of 70:30 one can assume random X-inactivation, only at more extreme ratios one can speak of skewed. However, I am not sure if there is a precise definition. This should be defined precisely. A ratio of exactly 50:50 should be the most frequent with an assumed normal distribution, but all possible intermediate states are conceivable.
Response: We thank you for this note. We have now modified, better explaining this crucial point.
- The legend for Figure 1 describes the gray and white cells in a wrong way. It should state: “(gray – Xma)” instead of “(gray – Xmi)”.
Response: Thanks for signalling us this point. We have now corrected it.
- The sentence “Cellular mosaicism is an advantage for females in X-linked diseases, because XCI is usually random and results in the presence of wild-type alleles in at least 50% of cells. In this way, wild-type gene expression allows female carriers to show no clinical symptoms.” Is provided a bit early. If I am not mistaken, this should be the result of the meta-analysis, I guess. Furthermore, I think it is a popular theory and a very good evidence-based explanation for many findings in other X-linked diseases as well. Some relevant references should be brought here to underline the significance of the topic. And the reason why 50% "activity" of a given gene is sufficient for freedom of symptoms should also be addressed, namely the assumed redundancy of the system.
Response: We have accepted your suggestion, moving and discussing this part in the conclusion.
- The section 3.2. Skewed XCI in asymptomatic females deals with whole population data, not with patients with diagnosed diseases or carriers whatsoever, correct? I think it should be named “Skewed inactivation in the normal (or healthy) population”. Asymptomatic is not healthy, It refers to a more specific cohort than just randomized “whole population data”.
Response: Thank you for the suggestion. We modified the title of the section as suggested.
- Lines 225-27: “…in fact familiar cases of skewed XCI have been reported, leading to severe clinical symptoms or discordant phenotype.” This is in my opinion contradictory to the sentence in lines 266/67: “The XCI pattern seems not to be inherited, because the mother-daughters did not show the same pattern in the carriers of dystrophinopathies…” For the latter statement two references are provided so I believe you can find evidence for both statements. As it is presented now, I find it contradictory.
Response: We have now modified the text, to better clarify this point.
- I have some concerns with the decision tree for literature selection for the meta-analysis. 3 articles are very few. Why were only studies with MSSI included? Why were case reports excluded? There are sometimes very interesting conclusions of the treating physicians that could be controversial. I only had a quick look at the literature, but Morrone et al (J Med Genet. 2003 Aug;40(8):e103.) describe 4 female patients, two are asymptomatic (random X-inactivation) and 2 manifest disease symptoms (skewed in favour of mutant chromosome). These are clear indications that this plays an important role. Of course, no clear statistics can then be superimposed on the data, but statistical significance is not everything, sometimes medical relevance is nevertheless given. As in the example of the inheritance of skewedness. It may not be the rule, but it can happen. However, it can be lost in the totality.
Response: we agree with this observation. However, we decided to exclude the small studies because a higher risk of bias. We selected the studies in which the phenotype was clearly reported. The MSSI scale was chosen because it is an instrument to quantify the severity of the disease. Cardiac phenotype was also included in the meta-analysis because it is the more frequent and important involvement in female Fabry carriers.
- The original studies (Dobrovolny et al., 2005; Maier et al., 2006; Echeivarra et al., 2015) should be better introduced. Table 1 is helpful, of course, but it would be important to know what limit was used to assess skewedness (70:30, 80:20) in the individual studies and were these adopted or did they have to be adjusted for the meta-analysis? Or were the actual values used instead of a binary yes/no?
Response: We accepted this point, so we have included it in the meta-analysis paragraph.
- One important question for me to fully understand the key message here: Do the original 3 studies individually claim that skewedness is significant for the phenotype and you falsified that? The authors of the original studies should also have come to the conclusion that leukocytes are unsuitable for an analysis of XCI?
Response: We thank the reviewer for the suggestion, we have now included this critical point in the meta-analysis paragraph.

Round 2
Reviewer 1 Report
This is a review and complete and large access to references should be provided
Author Response
We thank the reviewer.
Reviewer 2 Report
Dear authors,
I thank you for the careful revision of the manuscript and the consideration of my concerns in particular. I think the article has gained in regards of coherence and clarity. Especially the meta-analysis is now much more comprehensible and thus suitable for broad attention by the community. Still there are some few minor points left, but I would rather like to call it a final edit than a revision.
- References missing for important statements, e.g.: a) “The late-onset atypical forms are usually more benign, because they involve a more restricted number of organs, usually limited to the kidneys, heart, or nervous system.” (page 3/19, ll. 110-111) b) “Sometimes the cardiomyopathy in Fabry patients mimics the hypertrophic cardiomyopathy due to sarcomeric genes mutations, particularly when isolated. However, the global ejection fraction is preserved.” (p. 4/19, ll. 118-120) As a non-cardiologist, I would like to know which sarcomeric gene mutations are involved here and with which diseases, not only symptoms, they are associated. In terms of differential diagnosis, this could be important information that should also be referenced.
- Line 114: “heterozygous” female instead of hemizygote female. A hemizygous female would have lost an X-chromosome. I think this is not meant?!
- Line 332: MSSI instead of MMSI.
- Line 350: 7_presented (space missing)
In general, the text would become even more readable with a language proofread. As far as I can tell, however, the use of English is largely fine.
Author Response
We are very grateful to the reviewer for his observations. Below are the answers point by point.
- References missing for important statements, e.g.: a) “The late-onset atypical forms are usually more benign, because they involve a more restricted number of organs, usually limited to the kidneys, heart, or nervous system.” (page 3/19, ll. 110-111)
Response: We thank the reviewer for the suggestion, we have now added the references.
- b) “Sometimes the cardiomyopathy in Fabry patients mimics the hypertrophic cardiomyopathy due to sarcomeric genes mutations, particularly when isolated. However, the global ejection fraction is preserved.” (p. 4/19, ll. 118-120) As a non-cardiologist, I would like to know which sarcomeric gene mutations are involved here and with which diseases, not only symptoms, they are associated. In terms of differential diagnosis, this could be important information that should also be referenced.
Response: We have now inserted this point in the text, as suggested.
- Line 114: “heterozygous” female instead of hemizygote female. A hemizygous female would have lost an X-chromosome. I think this is not meant?!
Response: We thank the reviewer for this observation, we have now modified as suggested.
- Line 332: MSSI instead of MMSI.
Response: We have now corrected.
- Line 350: 7_presented (space missing)
Response: We have now corrected.

Round 3
Reviewer 1 Report
I might have not been clear enough and I am sorry about that. My point of view is that a review should provide exhaustive references. The authors thank me but they do not modify their text. therefore I cannot accept the review in the present form
Author Response
According to your suggestion, we have now added all the references indicated for the paragraph 1.5 - Treatment section.
Moreover, we have expanded the number of references related to the paragraph 1.3, Clinical presentation.
The review now counts n. 148 references (instead of 111 of the first version presented at MDPI), in line with the most part of reviews published in the “Genes” journal.
